# Association between the Potential Influence of a Lifestyle Intervention in Older Individuals with Excess Weight and Metabolic Syndrome on Untreated Household Cohabitants and Their Family Support: The PREDIMED-Plus Study

**DOI:** 10.3390/nu12071975

**Published:** 2020-07-03

**Authors:** Josep Basora, Felipe Villalobos, Meritxell Pallejà-Millán, Nancy Babio, Albert Goday, Olga Castañer, Montserrat Fitó, María Dolores Zomeño, Xavier Pintó, Emilio Sacanella, Indira Paz-Graniel, Jordi Salas-Salvadó

**Affiliations:** 1Unitat de Suport a la Recerca Tarragona-Reus, Fundació Institut Universitari per a la recerca a l’Atenció Primària de Salut Jordi Gol i Gurina (IDIAPJGol), 43202 Reus, Spain; fvillalobos@idiapjgol.info (F.V.); mpalleja@idiapjgol.info (M.P.-M.); 2Universitat Rovira i Virgili, Departament de Bioquímica i Biotecnologia, Unitat de Nutrició Humana, 43201 Reus, Spain; indiradelsocorro.paz@urv.cat (I.P.-G.); jordi.salas@urv.cat (J.S.-S.); 3Consorcio CIBER, M.P. Fisiopatología de la Obesidad y Nutrición (CIBERObn), Instituto de Salud Carlos III (ISCIII), 28029 Madrid, Spain; agoday@parcdesalutmar.cat (A.G.); ocastaner@imim.es (O.C.); mfito@imim.es (M.F.); mzomeno@imim.es (M.D.Z.); xpinto@bellvitgehospital.cat (X.P.); esacane@clinic.cat (E.S.); 4Institut d’Investigació Sanitària Pere i Virgili (IISPV), 43204 Reus, Spain; 5Cardiovascular Risk and Nutrition Research Group, Hospital del Mar Medical Research Institute (IMIM), 08003 Barcelona, Spain; 6Lipid Unit, Department of Internal Medicine, Bellvitge Biomedical Research Institute (IDIBELL)-Hospital Universitari de Bellvitge, L’Hospitalet de Llobregat, 08908 Barcelona, Spain; 7Department of Internal Medicine, Hospital Clínic, Institut d’Investigacions Biomèdiques August Pi Sunyer (IDIBAPS), University of Barcelona, 08026 Barcelona, Spain; 8Hospital Universitari Sant Joan de Reus (HUSJR), 43204 Reus, Spain

**Keywords:** overweight, obesity, metabolic syndrome, Mediterranean diet, healthy lifestyles

## Abstract

This cross-sectional study aims to evaluate the association between the PREDIMED-Plus study lifestyle intervention and (i) adherence to the Mediterranean diet (MedDiet) and (ii) physical activity of cohabiting study participants, and to define the related social characteristics of the household members. Participants were a subsample of 541 cohabitants of the PREDIMED-Plus study. Adherence to the MedDiet, physical activity, anthropometric measurements, family function, and social support were assessed. Multiple linear regressions were applied to the data. Partners of the PREDIMED-Plus participants had higher adherence to the MedDiet compared to their sons/daughters (9.0 vs. 6.9 points). In comparison to partners with low adherence to the MedDiet, partners with high adherence were older, practiced more physical activity, ate more frequently with the PREDIMED-Plus participants, and had better family function (adaptability item). Compared to physically active partners, very active ones were older, more likely to be women, and had lower BMI and higher adherence to the MedDiet. In addition, they ate more frequently with the PREDIMED-Plus participants and had better family function. Using multiple lineal regressions, an increase in the adherence to the MedDiet of the PREDIMED-Plus participant, and better family function, were positively associated with their partner’s adherence to the MedDiet. The PREDIMED-Plus intervention showed a positive association with adherence to the MedDiet of the study participants’ partners. In addition, this association was influenced by the social characteristics of the household members.

## 1. Introduction

Obesity is a chronic, multifactorial disease that not only reduces the quality of life and life expectancy but also represents a considerable economic burden for public health systems. The design of strategies to lose weight, which is key to controlling the obesity pandemic and its comorbidities, constitutes a challenge for both health authorities and professionals [1]. The WHO recommends that overweight or obese adults with some comorbidity should lose 10% of their initial weight, starting with a lifestyle intervention as the main tool. Nevertheless, this strategy requires a considerable investment of time and resources to achieve permanent changes.

Since the period required for effective dietary intervention is lengthy, the expansion of its benefits to household members could optimize the healthcare team’s resources. This expansion, known in some publications as the halo effect, has been demonstrated in family members of individuals with morbid obesity who have undergone bariatric surgery [2,3,4,5,6]. It is, however, currently unknown whether less aggressive treatments for obesity, such as dietary or lifestyle interventions, may also produce this potential expansive influence of the intervention on the rest of the household. When the partners of individuals subjected to a weight-loss diet are involved, the effect of the diet is more noticeable compared to when the partners do not engage [7,8,9]. Therefore, household members could also condition intervention compliance since eating patterns tend to be similar, including adherence and maintenance of some eating habits [10,11,12]. In contrast, if the treated individuals are immersed in a highly unfavorable family environment, this could lead to greater difficulty in achieving dietary and lifestyle changes. In fact, such a situation could not only nullify the expansive benefit of the intervention but also make it difficult for the treated individuals to lose weight.

Household members, family function and environment, and social support could play essential roles in either improving or impairing the adherence to a lifestyle intervention program. Identifying family function and social support could, therefore, be helpful and favor a better response to the control and management of individuals who are overweight. Such an association has been studied mainly in other eating disorders such as anorexia and bulimia, but not in the treatment of obesity with behavior modification. For example, two studies conducted in Spain involving adolescents with anorexia and bulimia reported that there was an inverse relationship between family function and the Eating Disorders Inventory score [13,14].

We, therefore, hypothesize that greater adherence to the Mediterranean diet (MedDiet) and a higher level of physical activity of the participants from the PREDIMED-Plus study would be directly associated with the same lifestyle behaviors of their household cohabitants. In addition, greater adherence to the intervention would be associated with greater social support from their cohabitants.

This project is an ancillary study from the PREDIMED-Plus trial [15], an ongoing study, in which we have recently reported, after 12 months, the effectiveness of a lifestyle intervention consisting of a Mediterranean diet with energy restriction (erMedDiet) and promotion of physical activity in overweight/obese individuals with metabolic syndrome [16].

## 2. Material and Methods

### 2.1. Study Design

The PREDIMED-Plus study is an ongoing, multicenter, randomized, controlled, clinical trial conducted in Spain for primary cardiovascular prevention involving subjects between 55 and 80 years with overweight/obesity and metabolic syndrome. The participants were randomized into two groups: (a) an intensive lifestyle intervention based on an erMedDiet, physical activity promotion, and behavioral support (intervention group), or (b) recommendations to follow an energy-unrestricted MedDiet without any advice to increase physical activity within the context of usual healthcare (control group). The study protocol is detailed in http://predimedplus.com/, and the description of the cohort has been published elsewhere [15].

The present work is based on a cross-sectional analysis of household cohabitants of the PREDIMED-Plus study participants, who had at least one year of intervention at the time of the study.

### 2.2. Participants, Recruitment, and Randomization

For the present study, all the individuals who lived in the same home as the PREDIMED-Plus study participants in the following four centers were invited to participate: (a) Institut Hospital del Mar d’Investigacions Mèdiques (IMIM) in Barcelona, (b) Hospital Sant Joan-IISPV/Atenció Primària in Reus, (c) Atenció Primària Metro Sur-Departament d’Aterioesclorosis de I’Hospital de Bellvitge in Barcelona, and (d) Hospital Clinic of Barcelona.

The household PREDIMED-Plus cohabitants were categorized according to the respective intervention group of the PREDIMED-Plus study participants. All the individuals aged over 18 years who lived in the same home as the participant (partner, offspring, parents, siblings, and/or friends) were considered household cohabitants. The research team from each recruitment center explained the study to the volunteers, and written informed consent was obtained from all participants. The study protocol was approved by the Ethics Committee of each participating center. The ethical principles and good clinical practices contained in the Declaration of Helsinki were respected.

### 2.3. Sociodemographic and Anthropometric Variables

Age, sex, educational level, and individual medical history were obtained from face-to-face interviews. Weight (kg) and height (cm) were measured with light clothing in a standardized manner, and body mass index calculated [BMI = kg/m^2^].

### 2.4. Adherence to the Mediterranean Diet

Adherence to the MedDiet was determined by a previously validated questionnaire of 14 items (Mediterranean Diet Adherence Screener; MEDAS) [17]. The subjects were classified in tertiles of adherence to the MedDiet according to the score obtained (Q1: ≤7 -low-, Q2: 8–10 -medium, and Q3 >10 -high adherence-); cut-offs were based on the study by Martínez-González et al. [17]. The frequency per week in which the cohabitants shared mealtimes with the PREDIMED-Plus participant was also recorded.

### 2.5. Physical Activity

Physical activity was measured using the Minnesota Questionnaire validated for the Spanish population [18,19]. Intensity (light, moderate, or vigorous), frequency (days per week), and duration of physical activity (minutes per day) were registered. The intensity and frequency of each activity were used to calculate the intensity category in terms of METs/min/week. These values were obtained by multiplying the average energy expenditure (3.3 MET for walking, 4.0 MET for moderate intensity, and 8.0 MET for vigorous intensity) by min/week for each physical activity. The results of each category of activity intensity were summed to obtain the total physical activity. Based on total physical activity, participants were classified into two categories: active (≤2100 METs/min/week) and very active (>2100 METs/min/week). 

### 2.6. Social Characteristics of the Household Cohabitants

#### 2.6.1. Family Function

The Family APGAR test (adaptability, partnership, growth, affection, and resolve), validated in the Spanish population, was used to assess the qualitative measurement of the household cohabitants’ satisfaction with each of the five basic components of family function. The participants checked one of three choices, which are scored as follows: “Almost always” (2 points), “Some of the time” (1 point), or “Hardly ever” (0 points). The maximum total score is 10; the highest score suggests a strongly functional family [20,21].

#### 2.6.2. Social Support

Perceived social support, which means the extent to which the basic social needs of individuals are met through interaction with others, was assessed using the Duke-UNC-11 Functional Social Support questionnaire, validated in the Spanish population. This tool measures two subscales: confidential support (possibility of having someone to communicate with), and affective support (demonstrations of love, affection, and empathy). It consists of 14 items (confidential support items: 1, 2, 6, 7, 8, 9, and 10; and affective support items: 3, 4, 6, and 11), each of which ranges from 1 to 5. The maximum total score is 40; the highest suggests better social support of the individual [22,23].

### 2.7. PREDIMED-Plus Lifestyle Intervention 

The following variables were considered as the lifestyle intervention program: adherence to the MedDiet and physical activity of the PREDIMED-Plus participant at the end of the intervention.

### 2.8. Statistical Analysis

The data are presented as mean and standard deviation (SD) for continuous variables, or as a median and interquartile range [IR] for non-normally distributed data, and percentages and numbers for categorical variables. We used *t*-tests or ANOVA-tests for comparisons of continuous variables among groups. The Mann–Whitney U-test or the Kruskall–Wallis test was employed for the continuous variables that did not present a normal distribution according to the Kolmogorov–Smirnov test. Comparisons among groups for categorical variables were performed with the χ^2^ test and Fisher test when the expected frequencies were less than 5.

The association between the PREDIMED-Plus participants’ lifestyle (adherence to the MedDiet and physical activity) and cohabitant adherence to MedDiet was assessed by a multiple linear regression model adjusted by age (years), sex (man/woman), highest education level (university, high school, secondary school, or elementary school), hypertension (yes/no), dyslipidemia (yes/no), type 2 diabetes mellitus (yes/no), BMI (kg/m^2^), family function (score), social support (score), and time of follow-up (in years). In addition, the association between PREDIMED-Plus lifestyle and cohabitant physical activity was assessed using a multiple linear regression model adjusted by the aforementioned variables.

Statistical significance was set at *p*-value < 0.05. The statistical software “R 3.4.3” for Windows was employed.

## 3. Results

After recruiting the potential participants declared as household members at the study baseline, 541 household cohabitants from 477 PREDIMED-Plus participants were included (Figure 1).

Table 1 reports the general characteristics of the household cohabitants of the PREDIMED-Plus study participants according to their family categories. Partners of the PREDIMED-Plus participants, compared to their sons and daughters, had significantly higher scores of adherence to the MedDiet (9.0 (1.9) points vs. 6.9 (2.3) points). In addition, partners were older and more likely to be women, with dyslipidemia, presenting lower BMI, better family function, and lower confidential support scores. In addition, they ate more frequently with the corresponding PREDIMED-Plus participant, compared to their sons and daughters (*p*-value < 0.05).

### 3.1. Associations among the PREDIMED-Plus Participants’ Sons and Daughters

No significant associations were observed in the main variables of the study (adherence to the MedDiet and physical activity) in the case of sons and daughters (data presented as Appendix A).

### 3.2. Associations among the PREDIMED-Plus Participants’ Partners

Table 2 describes the general characteristics of the partners according to their tertiles of adherence to the MedDiet. Compared to the partners with low adherence to the MedDiet, those with high adherence were significantly older, practiced more physical activity, and ate a greater number of times per week with the PREDIMED-Plus participant. In addition, a considerable proportion of partners with high adherence to the MedDiet were almost always satisfied that they could turn to their family for help when something was troubling them (adaptability item).

Table 3 describes the general characteristics of the partners categorized as very active or active. Compared to active partners, very active partners were significantly older, more likely to be women, presented lower BMI, higher adherence to the MedDiet, and ate more frequently per week with the respective PREDIMED-Plus participant. In addition, a higher proportion of very active partners were almost always satisfied that they could turn to their family for help when something was troubling them (adaptability item).

Table 4 summarizes the association between the lifestyle of PREDIMED-Plus participants (adherence to the MedDiet and physical activity) and their partner’s adherence to the MedDiet. The multiple linear regression model showed that an increase in one point in the adherence to the MedDiet of the PREDIMED-Plus participants was associated with an increase of 0.43 points in their partner’s adherence to the MedDiet (*p*-value < 0.001). In addition, an increase in family function was associated with an increase in the partner’s adherence to the MedDiet.

No significant associations were found between the lifestyle of PREDIMED-Plus participants and their partners’ physical activity (Table 5).

## 4. Discussion

In this cross-sectional study involving household cohabitants of the PREDIMED-Plus study participants, adherence to the MedDiet in participants with overweight/obesity following a lifestyle intervention was positively associated with their partners’ adherence to the MedDiet. This relationship was linked to household support, in particular, family function. We did not find, however, a significant association between the physical activity of PREDIMED-Plus participants and that of their partners.

There is limited knowledge regarding the relationship between adherence to a lifestyle intervention program and its potential expansive benefits on household cohabitants, and how family function and social support are related to such an association.

Previous studies have demonstrated a positive association between a MedDiet or low-fat one and the eating behaviors of household members [8,24,25,26]. A study conducted with the wives of men intervened with a MedDiet demonstrated that the wives experienced a decrease in the consumption of saturated fat, processed meat, pasta, pastries and sugar drinks, and an increase in the consumption of fish and vegetables [8]. Similar observations were found in other studies conducted in husbands of women intervened with a low-fat diet, showing an association between the level of fat consumed between the intervened women and their respective husbands [24,25,26]. Our results are in line with these results, suggesting a potential association of adherence to the lifestyle intervention with other untreated household members.

With respect to our results regarding family function and social support, prior research has described inconsistent results related to the association between these characteristics and eating behaviors. In a study conducted in individuals with diabetes, better family function was associated with higher adherence to a healthy diet [27]. In addition, results from two studies conducted in adolescents diagnosed with eating disorders reported a positive relationship between family dysfunction and unhealthy eating behaviors [13,14]. In contrast, another study observed that family function did not moderate or confound the associations between parental food practices and children’s nutrition risks [28].

The possible mechanisms underlying the association between the trial participants and their partners regarding adherence to the MedDiet could be explained by the fact that eating behavior is strongly influenced by social context. Food choices tend to converge with those of our close social nucleus, in this case, household cohabitants [29]. In addition, one of the reasons intervened individuals with healthy eating patterns exert an influence on the eating behavior of others is that they provide a guide or “social norm” for proper eating behavior [30]. Social norms are implicit codes of conduct that set an example for appropriate action. They may affect food choice and intake by altering self-perception and/or changing the sensory/hedonic evaluation of foods [29,30].

With respect to family function, it could play a role in the process of changing behaviors by acting as a moderator, potentially diminishing the effects of stressors found in lifestyle changes [31]. Specifically, positive family function, including healthy communication, greater family meal participation, and the presence of rules, structure, and problem-solving skills, could be protective for household members regarding good adherence to healthy dietary patterns.

Some limitations deserve to be mentioned, such as the inherent nature of cross-sectional studies that do not allow causality to be addressed. Moreover, our results cannot be extrapolated to other populations. It is important to highlight that although household cohabitants were more adherent to the MedDiet, they also had better family function. This could be a potential confounder since it is likely that participants with better family function would be more adherent to any particular healthy dietary pattern, rather than the intervention diet itself. Another limitation could be the absence of social support instruments whose psychometric properties have been thoroughly evaluated for use in the field of diet and excess weight. Nevertheless, the APGAR family test and Duke-UNC-11 questionnaire have been validated and employed in other studies conducted in the Spanish population, assessing social support for diet and eating disorders [13,14]. Finally, our study was conducted in Spain with a senior population displaying relatively high adherence to the MedDiet; our results cannot, therefore, be generalized to other younger populations not following healthy diet recommendations or with different family traditions and habits.

Our findings provide additional justification for conducting lifestyle programs in individuals with overweight and obesity. The benefits of the intervention program could be extended to the untreated partners of participants included in PREDIMED-Plus and represent a cost-effective method for changing the lifestyle for both. In addition, these results could assist in the design of lifestyle field interventions, comprehensively taking into account the social support provided by household members, and help strengthen intraconvivial relationships.

## 5. Conclusions

In this cross-sectional study involving the household cohabitants of the PREDIMED-Plus study participants, adherence to the Mediterranean diet in participants with overweight/obesity following a lifestyle intervention was associated with their partners’ adherence to MedDiet. This association could be mediated by the household members’ support, in particular, family function. We did not find, however, significant associations between the physical activity of the PREDIMED-Plus participants and the physical activity of their partners. More prospective studies are warranted to confirm our results.

## Figures and Tables

**Figure 1 nutrients-12-01975-f001:**
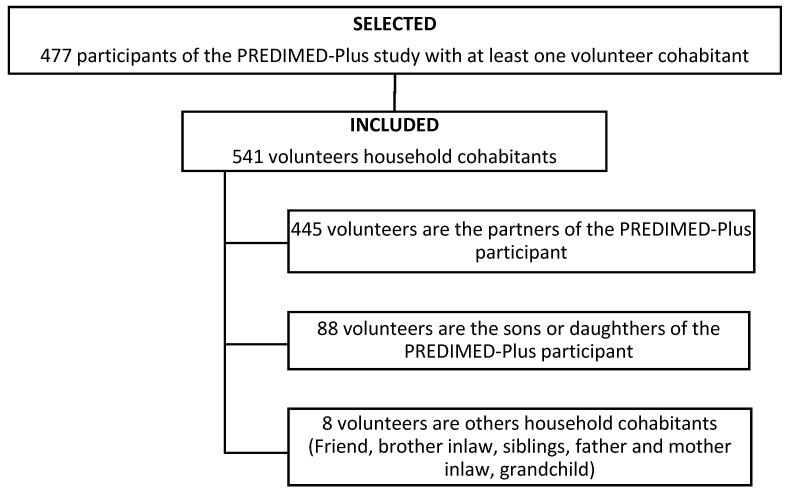
Flow chart of the study.

**Table 1 nutrients-12-01975-t001:** General characteristics of the household PREDIMED-Plus cohabitants.

	Son or Daughter *n* = 88	Partner *n* = 445	*p*-Value
**Sociodemographic characteristics**			
**Age** (**years**)	31.5 [25.0; 41.0]	66.0 [60.0; 70.0]	<0.001
**Sex ***			
Women	31.8 (28)	66.3 (295)	<0.001
Men	68.2 (60)	33.7 (150)	
**Education level ***			
University	20.5 (18)	17.4 (74)	0.080
High school	21.6 (19)	13.4 (57)	
Secondary school	25.0 (22)	23.0 (98)	
Elementary school	33.0 (29)	46.2 (197)	
**Anthropometric measures**			
Weight (kg)	78.6 (14.8)	74.1 (13.8)	0.030
Height (cm)	172 (9.05)	163 (7.95)	<0.001
BMI (kg/m^2^)	26.2 (5.0)	27.8 (4.5)	0.022
**Chronic disease prevalence ***			
Hypertension	29.9 (26)	40.4 (175)	0.085
Dyslipidemia	25.0 (22)	41.1 (177)	0.007
Type 2 diabetes mellitus	9.0 (8)	15.9 (69)	0.140
**Lifestyle**			
Adherence to the MedDiet (score)	7.00 [5.00; 8.00]	9.00 [8.00; 10.0]	<0.001
Physical activity (METs/min/week)	1954 [988; 3402]	1866 [871; 3432]	0.727
Eating together (times per week)	8.00 [6.00; 14.0]	16.0 [13.0; 21.0]	<0.001
**Social characteristics of the household PREDIMED-Plus cohabitants**			
Family function (score)	9.00 [8.00; 10.0]	10.0 [9.00; 10.0]	<0.001
Family APGAR items			
Adaptability *			0.875
Hardly ever	0.0 (0)	1.0 (3)	
Some of the time	12.5 (5)	15.5 (44)	
Almost always	87.5 (35)	83.4 (236)	
Partnership *			0.001
Hardly ever	0.0 (0)	2.4 (7)	
Some of the time	37.5 (15)	12.7 (36)	
Almost always	62.5 (25)	84.8 (240)	
Growth *			<0.001
Hardly ever	5.0 (2)	1.4 (4)	
Some of the time	35.0 (14)	11.0 (31)	
Almost always	60.0 (24)	87.6 (248)	
Affection *			0.123
Hardly ever	0.0 (0)	0.7 (2)	
Some of the time	32.5 (13)	18.7 (53)	
Almost always	67.5 (27)	80.6 (228)	
Resolve *			0.449
Hardly ever	0.0 (0)	0.3 (1)	
Some of the time	0.0 (0)	3.89 (11)	
Almost always	100.0 (40)	95.8 (271)	
Social support (score)	48.0 [44.2; 51.0]	48.0 [41.0; 51.0]	0.162
Social support sub-scales			
Affective support (score)	18.0 [16.0; 19.0]	18.0 [16.0; 20.0]	0.859
Confidential support (score)	31.0 [28.0; 33.0]	29.0 [25.0; 33.0]	0.031

Data are presented as mean (SD) and as median [IR] for continuous variables, and as % (*n*) for categorical variables *.

**Table 2 nutrients-12-01975-t002:** General characteristics of the household PREDIMED-Plus cohabitants according to Mediterranean diet adherence categories.

	Adherence to the MedDiet of the PREDIMED-Plus Participant			
	**≤7** ***n* = 48**	**8–10** ***n* = 137**	**>10** ***n* = 260**	***p-*** **overall**
Partner adherence to the MedDiet (score) ^‡^	7.21 (1.79) ^a^	8.55 (1.78) ^b^	9.56 (1.84) ^c^	<0.001
Partner physical activity (METs/min/week)	2355 (2542)	2522 (2873)	2724 (2357)	0.594
	**Partner adherence to the MedDiet (score)**			
	**≤7** ***n* = 101**	**8–10** ***n* = 164**	**>10** ***n* = 180**	***p*** **-overall**
**Sociodemographic characteristics**				
**Age** (**years**)**^‡‡^**	64.3 (7.3) ^a^	65.4 (7.6) ^b^	66.6 (6.65) ^c^	0.035
**Sex ***				
Women	65.3 (66)	64.0 (105)	68.9 (124)	0.618
Men	34.7 (35)	36.0 (59)	31.1 (56)	
**Education level ***				
University	20.2 (19)	13.8 (22)	19.2 (33)	0.202
High school	19.1 (18)	11.9 (19)	11.6 (20)	
Secondary school	22.3 (21)	26.9 43)	19.8 (34)	
Elementary school	38.3 (36)	47.5 (76)	49.4 (85)	
**Chronic disease prevalence ***				
Hypertension	42.4 (42)	42.4 (67)	37.5 (66)	0.592
Dyslipidemia	35.7 (35)	40.4 (63)	44.6 (79)	0.346
Type 2 diabetes mellitus	14.1 (14)	21.5 (34)	11.9 (21)	0.470
**Anthropometric measures**				
Weight (kg)	76.1 (14.4)	73.5 (13.7)	73.5 (13.5)	0.337
Height (cm)	163 (8.60)	163 (7.69)	163 (7.85)	0.997
BMI (kg/m^2^)	28.4 (5.0)	27.6 (4.7)	27.5 (3.9)	0.337
**Lifestyle**				
Physical activity (METs/min/week) **^‡‡‡^**	1,431 [626; 2240] ^a^	1,785 [852; 3043] ^b^	2,437 [1327; 3931] ^c^	<0.001
Eating together (times per week) **^‡‡‡‡^**	14.0 [10.0; 21.0] ^a^	14.0 [12.0; 21.0] ^b^	21.0 [14.0; 23.5] ^c^	0.001
**Social characteristics of the household PREDIMED-Plus cohabitants**				
Family function (score)	10.0 [9.00; 10.0]	10.0 [9.00; 10.0]	10.0 [9.00; 10.0]	0.100
Family APGAR items				
Adaptability *				0.031
Hardly ever	4.6 (3)	0.0 (0)	0.0 (0)	
Some of the time	15.6 (10)	19.6 (20)	12.0 (14)	
Almost always	79.7(51)	80.4 (82)	88.0 (103)	
Partnership *				0.172
Hardly ever	3.12 (2)	2.9 (3)	1.71 (2)	
Some of the time	14.1 (9)	6.8 (7)	17.1 (20)	
Almost always	82.8 (53)	90.2 (92)	81.2 (95)	
Growth *				0.980
Hardly ever	1.5 (1)	0.98 (1)	1.7 (2)	
Some of the time	12.5 (8)	10.8 (11)	10.3 (12)	
Almost always	85.9 (55)	88.2 (90)	88.0 (103)	
Affection *				0.138
Hardly ever	3.1 (2)	0.0 (0)	0.0 (0)	
Some of the time	23.4 (15)	16.7 (17)	17.9 (21)	
Almost always	73.4 (47)	83.3 (85)	82.1 (96)	
Resolve *				0.673
Hardly ever	1.5 (1)	0.0 (0)	0.0 (0)	
Some of the time	3.1 (2)	3.9 (4)	4.2 (5)	
Almost always	95.3 (61)	96.1 (98)	95.7 (112)	
Social support (score)	44.0 [39.0; 50.0]	48.0 [42.0; 51.0]	48.0 [41.0; 52.0]	0.084
Social support subscales				
Affective support (score)	17.0 [15.0; 19.0]	18.0 [16.0; 20.0]	18.0 [16.0; 20.0]	0.064
Confidential support (score)	27.0 [23.5; 32.0]	30.0 [25.0; 33.0]	29.0 [26.0; 33.0]	0.138

Data are presented as mean (SD) and as median [IR] for continuous variables, and as % (*n*) for categorical variables * *p*-value between-groups differences: ^‡^
*p*^ab^, *p*^bc^, and *p*^ac^ < 0.001; ^‡‡^
*p*^ac^ < 0.05; ^‡‡‡^
*p*^ab^ and *p*^bc^ < 0.05, *p*^ac^ < 0.001; ^‡‡‡‡^
*p*^ac^ and *p*^bc^ < 0.05.

**Table 3 nutrients-12-01975-t003:** General characteristics of the household PREDIMED-Plus cohabitants according to physical activity categories.

	Physical Activity of the PREDIMED-Plus Participant		
	**Very Active** ***n* = 289**	**Active** ***n* = 150**	***p-*** **value**
Partner adherence to the MedDiet (score)	9.10 (1.97)	8.80 (1.98)	0.137
Partner physical activity (Mets/Min/Week)	2705 (2628)	2411 (2147)	0.229
	**Partner physical activity**		
	**Very Active** ***n* = 187**	**Active** ***n* = 224**	***p*** **-value**
**Sociodemographic characteristics**			
**Age** (**years**)	66.6 (6.8)	65.0 (7.2)	0.029
**Sex ***			
Women	58.8 (110)	72.3 (162)	0.006
Men	41.2 (77)	27.7 (62)	
**Education level ***			
University	18.6 (33)	16.6 (36)	0.697
High school	11.3 (20)	15.2 (33)	
Secondary school	23.7 (42)	24.0 (52)	
Elementary school	46.3 (82)	44.2 (96)	
**Chronic disease prevalence ***			
Hypertension	40.1 (73)	38.4 (84)	0.798
Dyslipidemia	43.9 (79)	37.9 (83)	0.267
Type 2 diabetes mellitus	16.9 (31)	13.2 (29)	0.370
**Anthropometric measures**			
Weight (kg)	73.1 (13.0)	74.9 (14,5)	0.249
Height (cm)	163 (7.95)	162 (7.86)	0.207
BMI (kg/m^2^)	27.2 (3.9)	28.2 (5.0)	0.050
**Lifestyle**			
Adherence to the MedDiet (score)	10.0 [8.00; 11.0]	9.00 [7.00; 10.0]	<0.001
Eating together (times per week)	19.0 [14.0; 22.8]	14.0 [11.0; 21.0]	0.001
**Social characteristics of household PREDIMED-Plus cohabitants**			
Family function (score)	10.0 [9.00; 10.0]	10.0 [9.00; 10.0]	0.335
Family APGAR items			
Adaptability *			0.030
Hardly ever	0.0 (0)	2.1 (3)	
Some of the time	11.5 (15)	19.4 (27)	
Almost always	88.5 (115)	78.4 (109)	
Partnership *			1,000
Hardly ever	2.3 (3)	2.1 (3)	
Some of the time	13.1 (17)	12.9 (18)	
Almost always	84.6 (110)	84.9 (118)	
Growth *			0.334
Hardly ever	1.5 (2)	1.4 (2)	
Some of the time	12.3 (16)	7.1 (10)	
Almost always	86.2 (112)	91.4 (127)	
Affection *			0.602
Hardly ever	0.0 (0)	1.4 (2)	
Some of the time	19.2 (25)	18.0 (25)	
Almost always	80.8 (105)	80.6 (112)	
Resolve *			0.751
Hardly ever	0.0 (0)	0.7 (1)	
Some of the time	3.0 (4)	4.3 (6)	
Almost always	96.9 (126)	95.0 (132)	
Social support (score)	49.0 [42.0; 52.0]	47.0 [40.0; 51.0]	0.234
Social support subscales			
Affective support (score)	18.0 [16.0; 20.0]	18.0 [15.0; 19.0]	0.101
Confidential support (score)	29.0 [26.0; 33.0]	29.0 [25.0; 33.0]	0.350

Data are presented as mean (SD) and as median [IR] for continuous variables, and as % (*n*) for categorical variables *.

**Table 4 nutrients-12-01975-t004:** Associations between adherence to the Mediterranean diet of PREDIMED-PLUS participants and adherence to the Mediterranean diet of their household cohabitants.

Partner Adherence to MedDiet (Score)	β	SE	*p*-Value
Adherence to the MedDiet of the PREDIMED-Plus participant (score)	4.3 × 10^−1^	7.4 × 10^−1^	<0.001
Physical activity of the PREDIMED-Plus participant (METs/min/week)	−3.7 × 10^−5^	5.7 × 10^−5^	0.514
Age (years)	3.9 × 10^−2^	2.3 × 10^−2^	0.077
Sex (women)	3.6 × 10^−1^	3.2 × 10^−1^	0.257
Education level (university vs. high school)	−3.3 × 10^−1^	4.5 × 10^−1^	0.431
Education level (university vs. secondary school)	−3.7 × 10^−1^	3.7 × 10^−1^	0.316
Education level (university vs. elementary school)	−2.8 × 10^−1^	3.6 × 10^−1^	0.437
Hypertension (yes)	−2.0 × 10^−3^	3.1 × 10^−1^	0.994
Dyslipidemia (yes)	6.1 × 10^−1^	2.7 × 10^−1^	0.026
Type 2 diabetes mellitus (yes)	−2.0 × 10^−1^	3.7 × 10^−1^	0.594
BMI (kg/m^2^)	−2.9 × 10^−2^	2.9 × 10^−2^	0.322
Family function (score)	2.3 × 10^−1^	1.1 × 10^−1^	0.046
Social support (score)	1.2 × 10^−2^	1.9 × 10^−1^	0.508
Follow-up (years)	−2.5 × 10^−1^	3.1 × 10^−1^	0.415

Multiple linear regressions. Beta-coefficient and standard error are shown. Modelo: R^2^_C_ × 100 = 22%; F = 4.3; *p*-value < 0.001.

**Table 5 nutrients-12-01975-t005:** Associations between adherence to the physical activity of PREDIMED-PLUS participants and adherence to the physical activity of their household cohabitants.

Partner Physical Activity (METs/Min/Week)	β	SE	*p*-Value
Adherence to the MedDiet of thePREDIMED-Plus participant (score)	−5.1 × 10^1^	1.5 × 10^1^	0.657
Physical activity of the PREDIMED-Plus participant (METs/min/week)	1.6 × 10^−1^	8.8 × 10^−2^	0.064
Age (years)	3.0 × 10^1^	3.5 × 10^1^	0.398
Sex (women)	−8.1 × 10^1^	5.0 × 10^2^	0.103
Education level (university vs. high school)	−1.1 × 10^3^	7.4 × 10^2^	0.120
Education level (university vs. secondary school)	−1.1 × 10^3^	5.8 × 10^2^	0.070
Education level (university vs. elementary school)	−1.1 × 10^3^	5.6 × 10^2^	0.043
Hypertension (yes)	1.5 × 10^2^	5.0 × 10^2^	0.764
Dyslipidemia (yes)	8.8 × 10^2^	4.2 × 10^2^	0.039
Type 2 diabetes mellitus (yes)	−5.2 × 10^2^	6.1 × 10^2^	0.385
BMI (kg/m^2^)	−8.1 × 10^1^	4.6 × 10^1^	0.080
Family function (score)	6.4 × 10^1^	1.8 × 10^1^	0.720
Social support (score)	4.1 × 10^1^	2.9 × 10^1^	0.169
Follow-up (years)	−4.8 × 10^2^	4.9 × 10^2^	0.330

Multiple linear regressions. Beta-coefficient and standard error are shown. Modelo: R^2^_C_ × 100 = 8%; F = 1.9; *p*-value < 0.016.

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
