# Peer review of "Association between the Potential Influence of a Lifestyle Intervention in Older Individuals with Excess Weight and Metabolic Syndrome on Untreated Household Cohabitants and Their Family Support: The PREDIMED-Plus Study"

_nutrients, 2020, doi:10.3390/nu12071975_

Round 1
Reviewer 1 Report
Dear authors,
Thank you for taking an effort to report on the 'influence' of participants of a lifestyle intervention on the behaviour of family members. I do think it is an interesting topic. However, I do belief that this manuscript needs some serious improvements before it can be published. I sincerely hope that my feedback will be of use to you. Please, find my comments below. Because of time issues, I focus on things that need to be altered, not on issues that are okay. Best wishes!
A major methodological limitation is the fact that it is impossible to evaluate the effect of an intervention with a cross-sectional design. Hence, it is necessary to use the right terminology in title, abstract and paper to describe design and methods of this study in a truthful way. In addition, please don’t use the term ‘effect’ throughout the paper, but correlation instead.
In my opinion, the term ‘halo effect’ wasn’t used in the right context. The only ‘halo effect’ I am familiar with and is clearly defined in science (field of psychology), is a certain ‘cognitive bias’.
What do the authors mean with ‘convivial environment’? Convivial is used to describe the atmosphere of an environment and means ‘friendly environment’. This doesn’t seem the right definition for the environment the authors want to define.
The introduction lacks a clear rational of the relevance of the purpose of the current study including proper argumentation and a clear description of definitions used. That said, I do think it is a nice idea to explore lifestyle and interesting characteristics among household members of participants of a lifestyle intervention.
How can there be significant differences in APGAR scores when scores of partners and children seem to be almost equal? Of course, partners are older than their children!
Why did you analyze the differences in weight? That doesn’t seem to be meaningful without knowledge of height. Rather stick to analyzing differences in BMI.
The meaning of the text on lines 185-188 is not clear.
What and how did you exactly analyze the differences between tertiles of adherence (table 2)? Does the results in the table indicate significant differences between the three tertiles? Or did you also refine the analyses to discover which tertiles exactly differed from eachother (1 vs 2, 1 vs 3, 2 vs 3?)
What do you mean with ‘high adherence’ in the text (only >10 or both 8-10 and >10 combined)?
The footnote of table 4 indicates that the multiple regression model was adjusted for all variables included in the model. That is not the way how it is usually described. There are some relations of interest and the analyses of those variables are adjusted for the other variables in the model.
The authors start their discussion with the phrase ‘…a lifestyle intervention in patients with overweight or obesity is associated to changes in eating behaviours on people around them.’ However, you cannot conclude this from the current study that the intervention changed the behavior of partners and children.
Furthermore, the authors conclude that ‘This beneficial effect on the Mediterranean diet adherence, called HALO, is higher in those individuals with better family function.’. However, you cannot conclude this from the current study, since you did not analyze the effect on adherence nor the influence of family function on this relation (e.g. interaction effect).
With respect to tables 3 and 4, in my opinion, analyzing the difference between partners of participants with low adherence scores compared to partners of participants with high adherence scores would have given a better indication of the ‘influence of an intervention on other household members’ rather than exploring the differences between partners with low/high adherence scores.
Author Response
Dear Reviewer’s and Editor,
Thank you for your positive response to our manuscript (nutrients-770149) entitled: Association between the potential influence of a lifestyle intervention in older individuals with excess weight and metabolic syndrome on untreated household cohabitants and their family support: The PREDIMED-Plus study.
We have read all of the reviewer’s comments and we thank them for their constructive and helpful suggestions that have improved the manuscript (m/s). We have considered all the suggestions and have made the appropriate changes in the revised version of the text.
Also, we have had the manuscript systemically revised by a native English provider of editorial assistance. As such, we hope that all errors of grammar, style and have been eliminated.
We hope that the revised m/s has reached the quality requirements for inclusion in Nutrients.
Yours sincerely,
Corresponding Authors
Response Reviewer 1
Dear authors,
Thank you for taking an effort to report on the 'influence' of participants of a lifestyle intervention on the behaviour of family members. I do think it is an interesting topic. However, I do belief that this manuscript needs some serious improvements before it can be published. I sincerely hope that my feedback will be of use to you. Please, find my comments below. Because of time issues, I focus on things that need to be altered, not on issues that are okay. Best wishes! We sincerely thank to the reviewer the valuable comments and suggestions provided in the following lines, which have helped us to improve our m/s. We addressed all these comments in each of the following points, as well as in the manuscript when required.
Point 1. A major methodological limitation is the fact that it is impossible to evaluate the effect of an intervention with a cross-sectional design. Hence, it is necessary to use the right terminology in title, abstract and paper to describe design and methods of this study in a truthful way. In addition, please don’t use the term ‘effect’ throughout the paper, but correlation instead. Thank you for highlighting this important issue. In the new m/s we have included in the Discussion Section a limitation acknowledging that cross-sectional studies do not allow us to address causality. In addition, we have changed the term “effect” by “association” or “relationship” in the revised text. We believe that the term correlation is not appropriately used, and we have clarified this through all the m/s. We have changed the title of the paper accordingly: Association between the potential influence of a lifestyle intervention in older individuals with excess weight and metabolic syndrome on untreated household cohabitants and their family support: The PREDIMED-Plus study.
Point 2. In my opinion, the term ‘halo effect’ wasn’t used in the right context. The only ‘halo effect’ I am familiar with and is clearly defined in science (field of psychology), is a certain ‘cognitive bias’. To clarify the reader and as suggested by the reviewer, we have changed the “Halo effect” term by “potential expansive influence of the intervention on the rest of the household” (Page 4; line 113).
Point 3. What do the authors mean with ‘convivial environment’? Convivial is used to describe the atmosphere of an environment and means ‘friendly environment’. This doesn’t seem the right definition for the environment the authors want to define. This is the direct translation of a word very used in our culture to express what we would like to say. As suggested, we have changed the term “convivial environment” by the term “household cohabitant” or “household members”. This has been corrected in the revised text.
Point 4: The introduction lacks a clear rational of the relevance of the purpose of the current study including proper argumentation and a clear description of definitions used. That said, I do think it is a nice idea to explore lifestyle and interesting characteristics among household members of participants of a lifestyle intervention. Following reviewers’ suggestion, we rewrite the introduction (Page 4; lines 100 - 141).
Point 5: How can there be significant differences in APGAR scores when scores of partners and children seem to be almost equal? Of course, partners are older than their children! In the old m/s we express the medians, but when we explore differences, we used the means. This explain your questions. However, we believe that the best way to present the results regarding the APGAR items is as categorical variables to clarify the reader. This has been changed in table 1, 2 and 3 in the Results section of the revised text. As we have described in methodology, APGAR is constructed through 5 items, each item with the following answered: 'Almost always' (2 points), 'Some of the time' (1) point, or 'Hardly ever' (0), the score ranged from 0 to 2.
Point 6. Why did you analyze the differences in weight? That doesn’t seem to be meaningful without knowledge of height. Rather stick to analyzing differences in BMI. Thank you for highlighting this point. We collected the variable height. Therefore, we have included this variable in all the tables 1, 2 and 3.
Point 7. The meaning of the text on lines 185-188 is not clear. The sub-sample in this study involved the PREDIMED-Plus study participants’ household cohabitants. We observed that relatives’ categories (partners vs. sons and daughter) were not similar, significant differences were observed in many variables, including the main variables, and the number of sons and daughters were very low compared partners. According to that, we decided before to show all the analysis only for partners. However, this type of analysis has been clarified in the revised text and included as a supplementary material the analysis for sons and daughters. (Page 10; lines 262 - 265).
Point 8: What and how did you exactly analyze the differences between tertiles of adherence (table 2)? Does the results in the table indicate significant differences between the three tertiles? Or did you also refine the analyses to discover which tertiles exactly differed from each other (1 vs 2, 1 vs 3, 2 vs 3?) We have clarified this in tables of the new m/s. Now we show the p value for the comparison between the 3 categories (p-overal) but also we included the p-values for the comparisons between groups, showing only those significant values and making the appropriate changes in the table 2.
Point 9: What do you mean with ‘high adherence’ in the text (only >10 or both 8-10 and >10 combined)? We mean with high adherence a value >10 points. We have clarified and made the appropriate changes in the Methods Section of the revised m/s (Page 6; lines 182 - 183).
Point 10: The footnote of table 4 indicates that the multiple regression model was adjusted for all variables included in the model. That is not the way how it is usually described. There are some relations of interest and the analyses of those variables are adjusted for the other variables in the model. Thank you for this important comment. Following your comment and to clarify, we have changed the title of table 4 and the food note.
Point 11. The authors start their discussion with the phrase ‘…a lifestyle intervention in patients with overweight or obesity is associated to changes in eating behaviours on people around them.’ However, you cannot conclude this from the current study that the intervention changed the behavior of partners and children. We completely agree with the reviewer comment. Accordingly, we have changed the first paragraph of the discussion in order to clarify the most important findings of our study as follows.
“In this cross-sectional study involving household cohabitants of the PREDIMED-Plus study participants, adherence to the MedDiet in participants with overweight/obesity following a lifestyle intervention was positively associated with their partners’ adherence to the MedDiet. This relationship was linked to household support, in particular family function. We did not find, however, a significant association between the physical activity of PREDIMED-Plus participants and that of their partners”. (Page 16; lines 355-360).
Point 12. Furthermore, the authors conclude that ‘This beneficial effect on the Mediterranean diet adherence, called HALO, is higher in those individuals with better family function.’
However, you cannot conclude this from the current study, since you did not analyze the effect on adherence nor the influence of family function on this relation (e.g. interaction effect). We do agree with your observations. This has been corrected in the Conclusion section of the revised text. (Page 18; lines 424 - 430).
Point 13: With respect to tables 3 and 4, in my opinion, analyzing the difference between partners of participants with low adherence scores compared to partners of participants with high adherence scores would have given a better indication of the ‘influence of an intervention on other household members’ rather than exploring the differences between partners with low/high adherence scores. We have modified the tables’ according the reviewer comments.
Reviewer 2 Report
This study assessed the impact of the PREDIMED-Plus study on dietary adherence and physical activity among partners and spouses from the PREDIMED-Plus participants. The overall findings indicated that the partners of the study participants did have higher adherence to the Mediterranean diet than their sons/daughters.
Assessing adherence among participants’ family members is definitely an important topic for dietary interventions, since a lack of social support is a barrier for making healthier diet choices. However, I have a few suggestions to improve the manuscript:
Content:
- Although cohabitants were more adherent to the Mediterranean diet, it is noted that they had better family function scores. This could be a potential cofounder, since it is likely that participants with better family function would be more adherent to any particular healthy dietary pattern, rather than the intervention diet itself. I would add a sentence in the limitations section mentioning this.
- I know you had mentioned in the discussion that other studies also reported similar findings to yours, but do you think that cultural habits in Spain would make the family function scores (or generally, eating habits with family members) stronger or weaker than other countries? In other words, is adherence to the Mediterranean diet stronger in your sample because of family traditions and habits that are unique in Spain compared to other samples? It would worth adding this to the discussion.
- (Lines 54-55): A reference should be cited here to back this statement up.
Grammar/Formatting:
- References should be placed before the period/punctuation; e.g. [1], or [2,3].
- (Abstract, lines 30 and 36): In the abstract change “lineal” to linear regression.
- (Introduction line 46): Did you mean socio-economic? Socio-sanitary tends to get at things having to do with sanitary conditions like the water supply etc.
- (Tables 1 and 2): Some of the p-values have commas instead periods. It would be easier to read them if they were all consistent.
- (Line 187): Change “significantly” to significant.
- (Line 191): Change “with less weigh” to weighed less.
Author Response
Dear Reviewer,
We sincerely thank to the reviewer the valuable comments and suggestions provided in the following lines, which have helped us to improve our m/s. We addressed all these comments in each of the following points, as well as in the manuscript when required.
Also, we have had the manuscript systemically revised by a native English provider of editorial assistance. As such, we hope that all errors of grammar, style and have been eliminated.
We hope that the revised m/s has reached the quality requirements for inclusion in Nutrients
Yours sincerely,
Corresponding Authors
Response Reviewer 2
This study assessed the impact of the PREDIMED-Plus study on dietary adherence and physical activity among partners and spouses from the PREDIMED-Plus participants. The overall findings indicated that the partners of the study participants did have higher adherence to the Mediterranean diet than their sons/daughters.
Assessing adherence among participants’ family members is definitely an important topic for dietary interventions, since a lack of social support is a barrier for making healthier diet choices.
Thank you for the global appreciation of our m/s.
However, I have a few suggestions to improve the manuscript:
Content:
Point 1. Although cohabitants were more adherent to the Mediterranean diet, it is noted that they had better family function scores. This could be a potential cofounder, since it is likely that participants with better family function would be more adherent to any particular healthy dietary pattern, rather than the intervention diet itself. I would add a sentence in the limitations section mentioning this. We have discussed this issue in limitations part of the discussion Section as suggested.
“It is important to highlight that although household cohabitants were more adherent to the MedDiet, they had also better family function. This could be a potential confounder, since it is likely that participants with better family function would be more adherent to any particular healthy dietary pattern, rather than the intervention diet itself”. (Page 17; lines 403 - 406).
Point 2. I know you had mentioned in the discussion that other studies also reported similar findings to yours, but do you think that cultural habits in Spain would make the family function scores (or generally, eating habits with family members) stronger or weaker than other countries? In other words, is adherence to the Mediterranean diet stronger in your sample because of family traditions and habits that are unique in Spain compared to other samples? It would worth adding this to the discussion. This interesting question cannot be answered with the design of our study. There are some studies reporting that aged people and women have higher adherence to Mediterranean diet that young people and males. Our population is a senior population with a relatively high adherence to Mediterranean diet as previously reported in other PREDIMED papers. Probably this is because of family traditions and habits that are unique in Spain compared to other samples.
Even so, we have included the following sentence to the limitations of the Discussion Section: “Finally, our study was conducted in Spain with a senior population displaying relatively high adherence to the MedDiet, our results cannot therefore be generalized to other younger populations not following healthy diet recommendations or with different family traditions and habits”. (Page 18; lines 411 - 414).
Point 3. (Lines 54-55): A reference should be cited here to back this statement up. We have included a reference as suggested.
Point 4. Grammar/Formatting:
- References should be placed before the period/punctuation; e.g. [1], or [2,3]. Done, as suggested.
- (Abstract, lines 30 and 36): In the abstract change “lineal” to linear regression. This has been corrected in the new m/s (Page 3; lines 71).
- (Introduction line 46): Did you mean socio-economic? Socio-sanitary tends to get at things having to do with sanitary conditions like the water supply etc. We have changed the term “socio-sanitary” by “health-system” (Page 4; lines 101).
- (Tables 1 and 2): Some of the p-values have commas instead periods. It would be easier to read them if they were all consistent. Done, as suggested through all the m/s.
- (Line 187): Change “significantly” to significant. Done.
- (Line 191): Change “with less weigh” to weighed less. Done.
All errors of grammar/formatting mention above have been corrected in the revised text.